# Breakfast Skipping, Weight, Cardiometabolic Risk, and Nutrition Quality in Children and Adolescents: A Systematic Review of Randomized Controlled and Intervention Longitudinal Trials

**DOI:** 10.3390/nu13103331

**Published:** 2021-09-23

**Authors:** Roberta Ricotti, Marina Caputo, Alice Monzani, Stella Pigni, Valentina Antoniotti, Simonetta Bellone, Flavia Prodam

**Affiliations:** 1SCDU Pediatrics, Department of Health Sciences, University of Piemonte Orientale, 28100 Novara, Italy; roberta.ricotti@uniupo.it (R.R.); marina.caputo@uniupo.it (M.C.); alice.monzani@gmail.com (A.M.); antoniotti.valentina@gmail.com (V.A.); simonetta.bellone@med.uniupo.it (S.B.); 2Department of Health Sciences, University of Piemonte Orientale, 28100 Novara, Italy; stellapigni@libero.it; 3SCDU Endocrinology, Department of Translational Medicine, Università del Piemonte Orientale, 28100 Novara, Italy

**Keywords:** children, adolescents, breakfast skipping, obesity, nutrition, intervention, education, trial

## Abstract

Breakfast skipping increases with age, and an association with a high risk of being overweight (OW) and of obesity (OB), cardiometabolic risk, and unhealthy diet regimen has been demonstrated in observational studies with children and adults. Short-term intervention trials in adults reported conflicting results. The purpose of this systematic review was to summarize the association of breakfast skipping with body weight, metabolic features, and nutrition quality in the groups of young people that underwent randomized controlled (RCT) or intervention longitudinal trials lasting more than two months. We searched relevant databases (2000–2021) and identified 584 articles, of which 16 were suitable for inclusion. Overall, 50,066 children and adolescents were included. No studies analyzed cardiometabolic features. Interventions were efficacious in reducing breakfast skipping prevalence when multi-level approaches were used. Two longitudinal studies reported a high prevalence of OW/OB in breakfast skippers, whereas RCTs had negligible effects. Ten studies reported a lower-quality dietary intake in breakfast skippers. This review provides insight into the fact that breakfast skipping is a modifiable marker of the risk of OW/OB and unhealthy nutritional habits in children and adolescents. Further long-term multi-level intervention studies are needed to investigate the relationship between breakfast, nutrition quality, chronotypes, and cardiometabolic risk in youths.

## 1. Introduction

Childhood overweight (OW) and obesity (OB) are major public health issues in both developed and developing countries across the world. The prevalence of obesity has increased worldwide in children and adolescents from 1975 to 2016. By 2000, the trend in children’s and adolescents’ Body Mass Index (BMI) had reached a high-level-plateau in many high-income countries, but it is continuing to increase in East, South, and South-East Asia [1]. Several factors are involved in the obesity trend, such as genetic implications, birth weight, breastfeeding, parental obesity, physical activity, socioeconomic status, age, and gender [2]. However, dietary habits are among the main actors [3]; in particular, breakfast, the first meal of the day, has a critical role in energy balance and dietary regulation [4]. Weight imbalances during the first periods of life and across adolescence are associated with adverse health consequences over the course of life, particularly non-communicable diseases. Recent reviews identified a link between obesity and individual biological rhythm and feeding time. Inconsistent and irregular timing of food intake is associated with increased body weight [5]. Chronobiological studies investigating the rhythmic change of vital phenomena over time are clarifying the metabolic and circadian background of obesity [6]. Studies focused on the disruption of biological rhythms and relationships between feeding time, nutrition, circadian regulation, and metabolism reported that the time of food intake is an important determinant in the regulation of body weight [7]. Apart from dietary composition, the frequency and timing of meals also have an impact on these biomarkers, through regulation of the circadian clock and clock-controlled genes, modulation of satiety hormones and insulin levels, and lipid metabolism [8,9]. In our previous systematic review on the effects of breakfast skipping on weight and cardiometabolic risks factors among 286,804 children and adolescents recruited in cross-sectional studies, we demonstrated that breakfast skipping was associated with OW/OB in 94.7% of the subjects as well as being associated with a worse lipid profile, increased blood pressure levels, insulin-resistance, and metabolic syndrome (MetS) [10]. However, reviews in adults on longitudinal studies failed to demonstrate that the eating of breakfast results in weight loss [11]. The aim of the present systematic review was to analyze the association of breakfast skipping with body weight, metabolic outcomes, and nutrition quality in children and adolescents, focusing on interventional studies. Randomized controlled trials (RCTs), followed by longitudinal studies, are the gold standard for evidence-based medicine and are an appropriate tool for identifying a causal relationship between a specific nutrient or diet and a given health outcome in humans [12].

## 2. Materials and Methods

### 2.1. Literature Search

The study protocol was organized according to PRISMA-P guidelines [13], and the resulting report was written according to the Preferred Reporting Items for Systematic Reviews and Meta-Analyses (PRISMA) statement [14]. The PICO methodology (Population: pediatric population; Intervention: education to breakfast; Comparison: not education; Outcomes: weight and metabolic parameters) was used. We searched the Cochrane Central Register of Controlled Trials, PubMed, CINHAHL, and the EMBASE databases for the period 2000–2021. The reference lists of identified studies were also searched for all randomized and non-randomized clinical trials assessing the effects of promotion of breakfast consumption on body weight and cardiometabolic aspects in children and adolescents. No country restrictions were imposed. The search terms used included “breakfast”, “intervention”, “education”, “pediatrics”, “children”, “adolescents”. The search strategy used both keywords and MeSH terms. No further limitations were made so that the search terms would be as sensitive as possible. Only articles written in English were considered.

### 2.2. Outcome Measures

The primary outcome measures were: (1) body weight, BMI; (2) MetS, arterial hypertension, lipid profile, glucose levels, insulin resistance; and (3) nutrition quality. Included studies had to report at least one of these primary outcomes.

### 2.3. Inclusion and Exclusion Criteria

For inclusion, studies were required to (i) include children and/or adolescents aged 2–18 years (or a mean within these ranges) as subjects of study; (ii) have a defined measure of the child’s or adolescent’s breakfast consumption and/or breakfast skipping; (iii) be published in peer-reviewed journals in the English language; (vii) be published in the period 2000–2021; (viii) have a follow-up time of at least 2 months. We excluded studies if intervention was not described or if breakfast skipping was not defined.

### 2.4. Identification of Relevant Studies

Potentially relevant papers were selected by reading the titles and abstracts. If abstracts were not available or did not provide enough results, the entire article was retrieved and screened to determine whether it met the inclusion criteria.

### 2.5. Data Extraction and Synthesis

A form was generated to register whether individual studies met eligibility criteria and to collect data regarding the study design and methodological quality. Two authors (RR and AM) independently reviewed the titles and the abstracts in order to identify studies to be included in the review. Any disagreement was discussed with another reviewer (FP). Finally, the full texts of articles that passed the screening and eligibility steps were retrieved and read by the same reviewers. Any difference in opinion about the studies was resolved by discussion between all the investigators. The following data were extracted: author, publication year, study design, characteristics of the participants, description of the intervention, breakfast skipping definition, assessment methodology and reliability and validity of dietary measures, length of follow-up, methods of evaluation of body weight clinical outcomes and changes in cardiometabolic parameters, and nutrition quality.

### 2.6. Quality Assessment

The methodological quality of the included studies was assessed by two reviewers independently (MC and SP). The risk of bias within the included longitudinal studies was assessed by applying the ROBINS-I (“Risk Of Bias In Non-randomized Studies of Interventions”) tool [15], which covers seven domains: bias due to confounding; bias in selection of participants into the study; bias in classification of interventions; bias due to deviations from intended intervention; bias due to missing data; bias in measurement of outcomes; and bias in selection of the reported result. The categories for domain-level as well as for the overall risk of bias judgements were: low risk, moderate risk, serious risk, critical risk of bias, and no information. The study was classified as low risk if it was judged to be at low risk of bias for all domains, and as moderate risk if it was considered at low or moderate risk of bias for all domains. If the study was judged to be at serious or critical risk in any domain, it was classified as having a serious or critical risk of bias, respectively. “No information” was used only when there were insufficient data to permit judgement [15]. The risk of bias within the included RTCs was assessed using the Cochrane’s revised risk of bias tool RoB 2 [16]. The following domains were evaluated: bias arising from the randomization process; bias due to deviations from intended interventions; bias due to missing outcome data; bias in measurement of the outcome; and bias in selection of the reported result. Each domain was judged as recommended: low risk, high risk, or some concerns [16]. The study was classified as low risk if a low risk of bias for all domains was demonstrated, and as having a high risk of bias if high risk of bias in at least one domain or concerns in multiple domains lowering confidence in the result were demonstrated. Otherwise, if the study was judged to have some concerns in at least one domain but was not at high risk of bias for any domain, it was considered as having some concerns [16]. Consequently, the web app *robvis* was used to generate figures to present the risk of bias assessments for the selected studies [17]. Disagreements were resolved by discussion between the reviewers. Agreement between reviewers was good: K for agreement was 75.8% (k Cohen: 0.758) after screening titles and abstracts, and 100% after screening full-text articles.

## 3. Results

Our search identified a total of 584 potentially eligible studies. Of these, 441 were excluded, having been judged irrelevant on the basis of their titles. The remaining 143 records were screened by reading abstracts and a further 115 studies were excluded as they were observational studies with no intervention described. Thus, 28 articles were retrieved and underwent full-text assessment. Among these, 12 were excluded because they were observational studies without intervention (n = 7) or for reasons related to their outcomes (n = 1) or because breakfast skipping was not defined (n = 4). The selection process is summarized in Figure 1.

### 3.1. Study Characteristics

The literature search identified 16 potentially relevant articles. They are summarized in Table 1 and Table 2. Table 3 describes the interventions in detail. Of the 16 selected interventional studies, five were longitudinal studies [18,19,20,21,22] and 11 were RCTs [23,24,25,26,27,28,29,30,31,32,33]. Two papers referred to the same RCTs but reported different outcomes on the same population; the population was counted one time [31,32]. Interventions consisted of sessions of nutritional education in two studies [24,28], nutritional education messages and a school-breakfast program in six studies [21,22,25,27,29,33], free school-breakfast offered in six studies [18,20,23,26,31,32], free-home breakfast in one study [30], and a national breakfast promotion campaign in one study [19]. One educational program was used in a longitudinal study and afterwards in an RCT [21,29] (Table 3). Follow-up time recognized a wide range of variability from 2 months to 6 years, while in two studies follow-up time was not precisely specified [20,21]. Overall, data from a total of 50,066 subjects were reported. They came from eight different countries (Australia, Canada, Denmark, Egypt, Germany, Norway, USA, and Wales). Children’s age showed a wide range of variability from 1 year to 20 years. One study enrolled only preschool-aged children [24], 14 studies included school-aged children and/or adolescents [18,20,22,23,25,26,27,28,29,30,31,32,33], one study analyzed both school-aged and young subjects [19]. Some studies recorded data about breakfast skipping by food frequency questionnaires [18,23,24,26,28,29], some on a recall-based methodology or by food diaries [19,25,28,29,30,33], others with yes/no answers [19,20,22,25]. The definition of breakfast skipping was quite variable. Questionnaires were administered to the subjects or the parents in the case of youngsters. The subjects’ weight and height were measured only in some studies [19,23,25,28,30,33], and in the others were not reported [18,20,21,24,27,31,32]. To define OW and OB, BMI age- and sex-specific cut-offs according to international criteria were used [19,22,23,25,26,28,29,33]. In one study, also, blood pressure was measured [28].

### 3.2. Risk of Bias within Longitudinal Studies

The risk of bias assessment for the included longitudinal studies is presented in Figure 2a,b. All studies [18,19,20,21,22] were judged to be at moderate risk both for bias due to confounding and for bias in measurement of the outcome, mainly due to some expected, but not serious, residual confounding and to self-reported measures, respectively. Two studies (40%) [18,20] were judged to be at moderate risk for bias due to deviation from intended intervention, particularly in terms of implementation or adherence. All studies [18,19,20,21,22] were considered at low risk for bias in classification of intervention and bias due to missing data. Thus, the overall risk of bias judgment for all the selected longitudinal studies was “Moderate risk”.

### 3.3. Risk of Bias within RCTs

The risk of bias assessment for the selected RCTs is presented in Figure 3a,b. Almost all studies (81.8%) [23,24,25,26,27,28,29,30,31] were judged to have some concerns with respect to one or more domains. In particular, seven studies (63.6%) [23,24,26,27,28,29,30] were classified as having some concerns with respect to bias arising from the randomization process, mainly due to lack of information regarding allocation concealment; six studies (54.5%) [23,25,26,27,28,29] were judged to have some concerns in bias with respect to measurement of the outcome, as data collection relied on self-reported questionnaires with a potential for bias; and six studies (54.5%) [23,24,26,28,30,31] had some concerns with respect to selection of the reported results due to the unavailability of a trial protocol/statistical analysis plan. All trials [23,24,25,26,27,28,29,30,31,32,33] were considered at low risk for bias due to deviations from intended interventions.

### 3.4. Findings

Regarding results after the interventions, eight studies described a positive effect with increased frequency in eating breakfast at follow-up [19,20,22,23,24,26,28,33], while no changes were reported in four studies [25,27,29,32] and a negative final impact was reported in one study [18]. However, the study of Kobel et al. reported an increase in breakfast skipping only in the control group, with stable prevalence in the intervention group, although the results were not significant [29]. Interestingly, one study reported a decrease in breakfast skipping if the intervention was related to a delay of the starting time of the school [22]. Two longitudinal studies and three RCTs reported data about BMI and its correlation with breakfast skipping and/or health promotion intervention [19,21,23,30,33]. O’Dea et al. described a greater prevalence of combined OW/OB in 2006, after a national breakfast promotion campaign, than in 2000 (24.8% versus 21.6%). Therefore, among all participants, those with OW or OB were more likely than normal-weight students to miss breakfast either at baseline and at follow-up [19]. Similarly, Traub et al. reported that children who skipped breakfast at baseline were significantly more likely to show increases in waist-to-height ratio, weight, and BMI, in addition to being more highly associated with abdominal obesity at follow-up [21]. Three RCTs did not report significant effects on weight or BMI but these trials were relatively short, with a range of follow-up from three to eight months [23,30,33]. None of the studies investigated glucose metabolism and MetS after the intervention. Only Elseifi et al. measured blood pressure without describing specific results [28]. The nutritional impact was investigated in 11 studies [19,20,23,25,26,28,29,30,31,32,33]. Two studies reported that the intervention group increased juice consumption and decreased consumption of sugar-sweetened beverages and foods high in saturated fat and added sugars, in comparison with those who did not participate in the school-breakfast program [24,29]. Therefore, the other authors [20,23,26,28,30,31,32] described the highest proportion of healthy breakfasting after the intervention. All the interventions regarding schools in low economic development areas reported an improvement in nutrition quality [19,25,31,32].

## 4. Discussion

Breakfast is the first meal of the day that ends the night fasting period and is deserving of special attention in the field of chrono-nutrition. Several observational studies of all ages reported an increased prevalence of OW/OB in people who skip breakfast [10,34,35,36,37,38], although these findings have been questioned in two recent meta-analyses on interventional studies in adults [11,39]. Our review demonstrates that breakfast skipping remains associated with a high prevalence of OW/OB and a worse nutritional profile in intervention RCTs and intervention longitudinal trials in children and adolescents.

Manipulation of caloric intake or meal timing in a healthy balanced diet can delay the onset and progression of non-communicable diseases in humans and increase longevity in several organisms [40,41,42]. Time-restricted feeding regimens suggest consuming daily foods in a limited interval of the day in the active phase without caloric restriction but promoting a longer fasting period. Such regimens show higher associations with improved cardiometabolic features and decreased weight than do control diets [43,44,45]. Breakfast skipping/eating is an important issue in this debate, particularly in pediatric nutritional education. In children, the morning chronotype is prevalent, with a shift toward eveningness during puberty and late adolescence for several quite unexplored physiological, social, and environmental reasons [46]. Because eating habits seem to be strictly connected with chronotype and progressively consolidate from childhood to adulthood, strategies to counteract circadian misalignment, unhealthy food regimens, and cardiometabolic disease risk are of importance for the general population.

We selected RCTs and dietary intervention longitudinal trials with the aim of understanding whether breakfast consumption, indirectly favoring late eating in the evening and a healthy food regimen, achieved the goal of lowering the prevalence of OW/OB and cardiometabolic risk, and curbing unhealthy nutrient patterns. Studies included a wide population of 50,066 children and adolescents living in eight countries (seven western countries and one African country), a quite good representation of food habits over a fairly large area of the world, but these numbers remain lower than those previously published in reviews on observational studies [10,34,35]. This aspect is linked to our restrictive inclusion criteria, but also to difficulties in performing trials with an impact on lifestyle. In fact, when children and adolescents are recruited, studies should involve a series of social determinants, such as parents, school, and peers [12,47]. In line with this point, certain of the studies considered involved parents/caregivers [24,27] or the school setting [18,19,20,21,25,27,28,31,32].

Four out of five longitudinal studies promoting breakfast eating reported the skipping habit outcome with conflicting results, with breakfast skipping decreasing in three of them [19,20,22] and increasing in the remaining one [18]. The difference between the studies could depend on several factors, including the time of observation and the local or national context of the study. In fact, the three studies that reported positive results were national or nation-wide campaigns [19,20,22], one of them involving the use of mass media, with additional nutrition advice given at school as part of a health-promoting program and six years of follow-up [19], whilst the study with negative results was a local, less structured program with only one year of follow-up [18]. Interestingly, one of the effective studies reported a decrease in breakfast skipping in schools that delayed their starting time by about 1 h, suggesting that more time in the morning could help maintain a morningness chronotype [22,46]. Similarly contrasting results were reported in the trials with five efficacious studies [23,24,26,28,33] and three others without changes in the percentage of breakfast skipping [25,27,32]. The studies reporting no increase in breakfast eating were all school breakfast policies with free or reduced-price meals at school, while the others were associated with an educational program on the role of breakfast on health. This division across all the studies suggests that well-rounded and prolonged efforts, involving educators, media, governments, policymakers, physicians, and peers, should be expected to have a significant impact on lifestyle habits. Economic discounts are useful but not enough, even where children from low-income families are concerned [48,49]. Furthermore, the wide variety of definitions of breakfast skipping could have affected the results of negative studies.

The two longitudinal studies aiming to evaluate weight reported that OW/OB prevalence after 1 or 6 years, in particular visceral adiposity, is higher in children and adolescents who skipped breakfast, and that this result is independent of age and socio-economic status [19,21]. Although relatively few, these findings confirm those reported in a number of observational studies in children and adolescents [10,34,35]. The authors suggested that this effect derives from circadian rhythms, eating more in the evening, length of night fasting, and lower physical activity levels. Interestingly, some short-term cross-over studies which we did not include in our review due to differences in aims reported that introducing a protein breakfast in adolescents who frequently skipped breakfast reduced food craving and brain activation in regions controlling food motivation and reward, and increased satiety through dopamine mechanisms. Thus, they postulated that strategies with palatable foods and dietary strategies, including breakfast and increased dietary proteins, could restore the blunted dopamine pathway associated with obesity [50,51,52]. Two recent meta-analyses on RCTs in adults reported that introducing breakfast or breakfast skipping has quite negligible effects on weight [11,39]. We found three RCTs with findings on weight in adolescents that had results similar to those for adults [23,30,33]. However, the effects observed in the intervention longitudinal studies we selected are in contrast with the results of RCTs in both children and adults. These differences could be due to several factors, such as age, physical activity, socioeconomic status, ethnicity/country, breakfast and daily nutrient quality, and the use of drugs or other modifiers (tobacco, alcohol, substitute meals, etc.). Furthermore, all the studies with adults (mean duration between 7 and 8.6 weeks) and two out of the three studies with adolescents (lasting 12 to 16 weeks) are quite short [23,30], and therefore bias could have been introduced. Long-term dietary weight loss intervention studies have shown that the weight behavior observed has two phases, with the first one, usually after 6 months, characterized by weight loss, and the second one by weight maintenance or regain [53,54]. Indeed, results in short-term RCTs could not be expandable due to long-term effects. The third RCT study in adolescents had an observation time of 8 months, but the authors concluded that the lack of positive effects on weight could have resulted from poor participation and a failure to reach a moderate-to-high intensity level regarding the interventions [33].

Unfortunately, none of the studies investigated metabolic alterations, in contrast to several observational studies [10]. Only one study evaluated blood pressure values, though not with respect to the intervention [28]. A recent meta-analysis on case-control, cross-sectional, longitudinal, or cohort studies in adulthood observed that breakfast skipping was associated with an increased risk of heart disease and with a pooled HR/OR of 1.24 [55]. Efforts are needed to provide further insight into the role of breakfast and its relation to cardiometabolic risk in children in order to help lifestyle prevention programs.

The most important result concerns the quality of diet. Eleven studies investigated nutrition quality and ten out of eleven reported that eating breakfast was associated with an improvement in dietary quality. A decreased intake of sugar-sweetened beverages, saturated fats, and added sugars, or generally unhealthy choices were reported [19,20,23,25,26,28,29,30,31,32]. It is interesting to note that Ritchie et al. recorded a higher breakfast quality in those who consumed it in the classroom compared to those who ate it in the school cafeteria, and Ask et al. achieved the same results with respect to education on breakfast at home [32], suggesting that, in addition to the nutrient composition of foods, environment and peers have a role that ought to be investigated in further studies [20]. Our data confirm that interventions aimed at increasing breakfast consumption are effective strategies for improving nutrition education and quality, and that this also applies for economically deprived schools [19,25,31,32]. This supports the hypothesis that overall dietary quality, among other socioeconomic or health-related behaviors, may be responsible for the protective effects against weight gain and cardiometabolic risk associated with breakfast consumption recorded in the observational studies and intervention trials analyzed here [10,19,21,23,26,29,30,34,35,36,37,38,55]. Health-promoting environments are further elements to consider in relation to protective effects concerning weight and metabolism [20,23].

This review has several limitations. In fact, we retrieved relatively few studies on the topics, and data on OW/OB are limited, whereas those on cardiometabolic features are absent. Furthermore, the heterogeneity in the design and the lack of information on daily food intake in most of the studies suggest that our findings ought to be interpreted with caution. However, restricting the analysis to RCTs and intervention longitudinal studies allowed for a causal interpretation of the results, reinforcing our previous findings on observational studies [10]. All the selected studies were at low (RCTs) or moderate risk (longitudinal studies) of bias, suggesting that a certain level of confidence can be placed in the main results. The primary risks are the result of self-reported data, very frequent in population studies, or a lack of detailed information on randomization or adherence. The latter point suggests putting more methodological effort into trials aiming to change lifestyle habits, considering that tailored adherence and empowerment approaches are critical [56].

With these limitations acknowledged, our systematic review on intervention studies suggests that breakfast skipping is associated with a high prevalence of OW/OB and a low nutrition quality—one of the potential players in weight gain and metabolic derangement (Figure 4). Further high quality and long-term randomized controlled and longitudinal trials are needed to evaluate whether children and adolescents who eat breakfast are better protected against OW/OB and cardiometabolic risk than those who skip it, and whether strategies to introduce breakfast eating in young people is efficacious in the prevention of circadian rhythm misalignment. Our findings provide insights into the need for a multi-level approach to impact on lifestyle habits.

## Figures and Tables

**Figure 1 nutrients-13-03331-f001:**
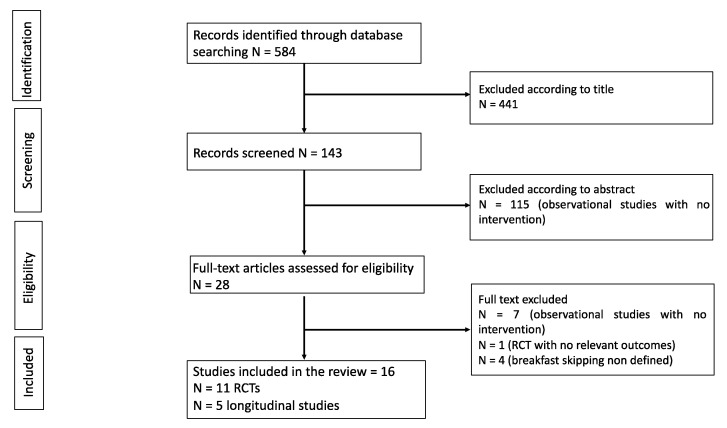
Flowchart of study selection.

**Figure 2 nutrients-13-03331-f002:**
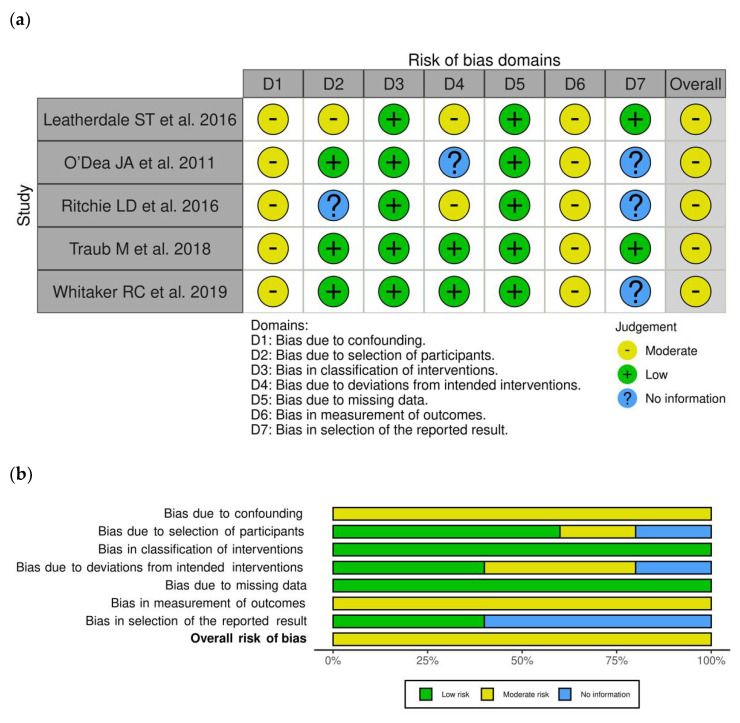
Risk of bias within longitudinal studies. (**a**) Traffic light plot and (**b**) summary plot presenting the risk of bias within the longitudinal studies included in the systematic review.

**Figure 3 nutrients-13-03331-f003:**
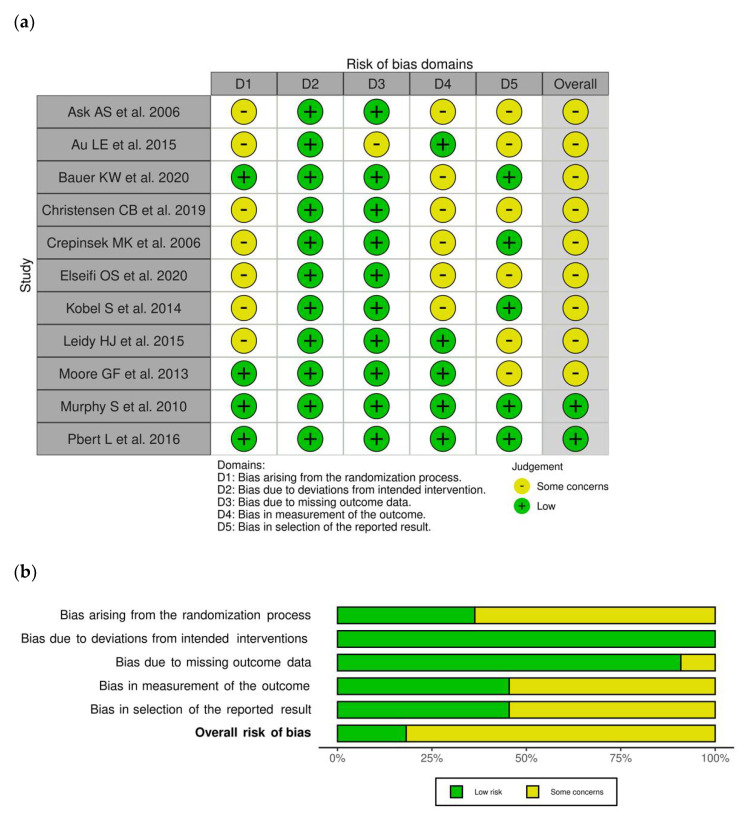
Risk of bias within RCTs. (**a**) Traffic light plot and (**b**) summary plot presenting the risk of bias within the RCTs included in the systematic review.

**Figure 4 nutrients-13-03331-f004:**
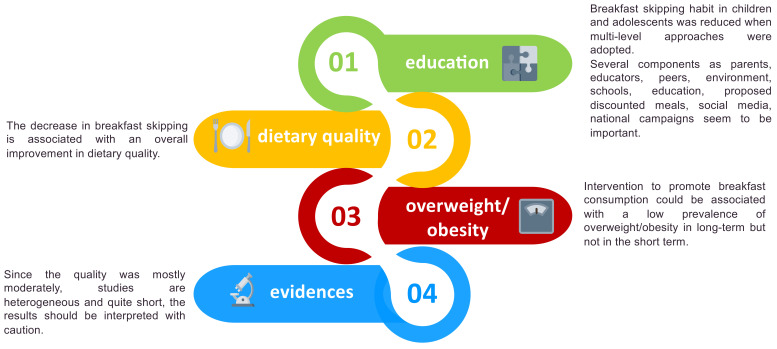
Key messages from the studies.

**Table 1 nutrients-13-03331-t001:** Characteristics and results of longitudinal studies included in the systematic review.

Author, Year, Reference	Subjects	Design	Intervention	Follow-Up Time	BS Definition	OW/OB Definition	General Results	BS	OW/OB	BP	Lipid Profile	Glucose	MetS	Nutrition Quality
Leatherdale ST, 2016 [18]	Secondary school students (grades 9–12th) T0: 23,921 T1: 23,117 M/F, Canada	COMPASS longitudinal Study	School breakfast program	1 year	Breakfast < 5 days in a usual school week (Monday to Friday)	NA	T0 Free school breakfast program: 37 secondary schools Paid school breakfast program: 1 secondary school No school breakfast program: 5 secondary schools T1 ↔ in school breakfast program: 38 secondary schools (control schools) Changed free school breakfast program: 3 secondary schools (intervention schools) Starting free school breakfast program: 1 secondary school (intervention school) Stopping free school breakfast program: 1 secondary school (intervention school)	T1 vs. T0 ↑ BS prevalence (54.5 vs. 54.9%) ↑ Breakfast program participation (12.3 vs. 13.6%)	NA	NA	NA	NA	NA	NA
O’Dea JA, 2011 [19]	School-children 7–18 yrs 32 schools T0: 4237 T1: 5645 M/F, Australia	Longitudinal study	National breakfast promotion campaign	2000 vs. 2006	“No” response to the question: “On most days, do you usually eat or drink something for breakfast? (yes/no)”	IOTF	T1 vs. T0 ↑ Prevalence of OW/OB: 24.8% vs. 21.6%; χ^2^= 13.78 ↔ prevalence of NW	T1 vs. T0 ↓ BS: - all students: 12.7 vs. 17.0%; χ^2^ = 35.12, *p* < 0.0001 - M: 11.6 vs. 15.3%; χ^2^ = 42.24, *p* < 0.0001 - F: 13.8 vs. 18.9%; χ^2^ = 22.77, *p* < 0.0001 - primary school: F: 9.6 vs. 6.0%; M: 9.4 vs. 6.0% - secondary school: F: 27.7 vs. 18.7%; M: 19.9 vs. 12.1%	T1 vs. T0 ↑ Prevalence of OW/OB : 24.8% vs. 21.6% BS among OW/OB > NW participants, both at T0 (20.7 vs. 16.0%, χ^2^ = 11.1) and T1 (14.3 vs. 10.4%)	NA	NA	NA	NA	T1 vs. T0 ↑ nutritional quality of breakfast scores - M: all year groups - F: mainly older year groups ↑ in students from low socio-economic status schools
Ritchie LD, 2016 [20]	3944 students (4–5th grade) in 43 low-resource elementary schools M/F, USA	Observational study; longitudinal baseline data of a cluster RCT	School-based intervention	NA	Zero calories recorded as breakfast	NA	/	Breakfast in the classroom: - ↓ BS students - ↑ eating breakfast at both home and school (*p* < 0.001)	NA	NA	NA	NA	NA	Breakfast in the classroom: ↑ overall dietary quality
Traub M, 2018 [21]	1733 children aged 7.08 ± 0.6 yrs M/F, Germany	Prospective, cluster randomized and longitudinal study	School-based health promotion program	1 year	Frequency of breakfast before school “often/always” vs. “never/rarely” → BS	OW: >90th BMI percentile OB: >97th percentile (German charts)	T1 BS among F > M	NA	BS ↑ odds of abdominal OB/OW at follow-up BS ↑ WHtR, weight, BMI percentiles, BMI (0.21 ± 0.01) and BMI z-scores (0.09 ± 0.03)	NA	NA	NA	NA	NA
Whitaker RC, 2019 [22]	3197 students (≥12 and <19.0 yrs) 19 schools (8 middle, 8 high, 3 secondary schools) M/F, USA	Quasi-experimental study	School start time changes	1 year	“No” response to the question: “On most school days, do you eat breakfast before your first class begins? (yes/no)”	NA	T1 vs. T0 ↓ BS in 50 min delay school	↓	NA	NA	NA	NA	NA	NA

**Legend:** BS= breakfast skipping; F = Females; M = Males; yrs = years; OB = obesity; OW = Overweight; NW = Normal Weight; BMI = Body Mass Index; BP = Blood Pressure; MetS = metabolic syndrome; WHtR = Waist-to-Height Ratio; IOTF = International Obesity Task Force; ↑ = Increased; ↓ = Reduced; ↔ No variation; NA = Not available. The last seven columns describe the impact of BS on those variables.

**Table 2 nutrients-13-03331-t002:** Characteristics and results of RCTs included in the systematic review.

Author, Year, Reference	Subjects	Design	Intervention	Follow-Up Time	BS Definition	OW/OB Definition	General Results	BS	OW/OB	BP	Lipid Profile	Glucose	MetS	Nutrition Quality
Ask AS, 2006 [23]	54 students; 15 yrs; 2 classes 10th grade secondary school M/F, Norway	Cn = 28 In = 26	Breakfast at school	4 months	Not having breakfast every day/week	IOTF	T1 vs. T0 In: ↑ breakfast every day; back to breakfast habits after 1 week	↓	T1 vs. T0 Cn: ↑ weight (*p* < 0.01) and BMI (*p* < 0.05) In: ↑weight in M (*p* < 0.05) ↔ OW/OB in both groups	NA	NA	NA	NA	T1 vs. T0 In: - ↑ healthy eating index in M (*p* < 0.01) - ↔ food supplements intake
Au LE, 2015 [24]	590 caregivers of 1–5-year-old children F, USA	Cn (In-person education) = 359 In (Online education) = 231	In-person vs. Online nutrition education	2–4 months	Breakfast eating frequency questions from the Healthy Kids Survey	NA	↓ barriers to eating breakfast in both groups ↑ frequency of eating breakfast in the online > in-person education group	↓	NA	NA	NA	NA	NA	NA
Bauer KW, 2020 [25]	1362 students 10.8 ± 1.0 yrs, family incomes ≤130% or 130–185% of the federal poverty level M/F, USA	Cn = 723 In = 639	Breakfast at school	2.5 years	“Did you eat/drink anything today?” “No” → BS	CDC	T1 vs. T0 In vs. Cn: ↑ breakfast consumption at school (44.3% vs. 13.1%)	↔	NA	NA	NA	NA	NA	T1 vs. T0 In vs. Cn: - ↑ fruit juice (25.4% vs. 14.7%); - ↓ sugar-sweetened beverages (10.6% vs. 15.6%) - ↓ foods high in saturated fat and added sugar (20.9% vs. 26.9%)
Christensen CB, 2019 [26]	318 students ≥ 16 yrs 4 vocational schools T0: 253 T1 (week 8): 168 T2 (week 14–16): 104 M/F, Denmark	Cluster-RCT 2: 2 schools Cn and In not correctly reported	Breakfast club intervention at school	4 months	Breakfast consumption (<9:00 a.m.) self-reported frequency scale; “daily breakfast eaters” if breakfast at school all days (yes/no)	NA	↑ daily breakfast eaters T1 vs. T0 - ↑ In vs. Cn (OR: 3.77) T2: NS	↓	NA	NA	NA	NA	NA	↑ WG products in In vs. Cn at T1 (OR: 4.13) and T2 (OR: 3.27)
Crepinsek MK, 2006 [27]	4278 primary school students 9.8 ± 1.3 yrs M/F, USA	Cn = 2066 In =2212	Free breakfast at school	1 year	Not having breakfast. “Any breakfast” = consumption of any food or beverage; or “a nutritional breakfast” = 2 of 5 food groups and energy > 10% 1989 REA	NA	T1 In vs. Cn - ↑ school breakfast - ↔ any breakfast eating - ↑ of a nutritional breakfast - ↔24h dietary intakes	↔ any breakfast - ↑ nutritional breakfast	NA	NA	NA	NA	NA	NA
Elseifi OS, 2020 [28]	Students 12–14 yrs M/F, Egypt	Pre-post In study Cn = 112 In =112	Nutritional education message	5 weeks + 2 months	Frequency breakfast per week, from 0 to 7 days: BS = 0–2 semi-BS = 3–4 non-BS = 5–7	WHO	↑ Breakfast in students with normal BMI and BP	T1 vs. T0 In: BS↓ 36.6% vs. 19%, non BS↑ 28.6% vs. 44.7	NA	NA	NA	NA	NA	T1 vs. T0 In: ↑ healthy breakfast BS: 57.1% vs. 68.8%, non-BS: ↑ 28.6% vs. 44.7
Kobel S, 2014 [29]	Primary school children 154 classes 7.1 ± 0.6 yrs T0: 1943 T1: 1736 M/F, Germany	Prospective, cluster RCT and longitudinal study Cn= 74 In= 80	School-based health promotion program	1 year	Frequency of breakfast before school “often/always” vs. “never/rarely” → BS	OW: >90th BMI percentile OB: >97th percentile (German charts)	T0 BS F > M (15.4 vs. 10.6 %) T1 - ↑ BS in Cn (NS); ↔ in In - BS among second-graders in Cn > in In (OR = 0.52, 95% CI 0.30; 0.92)	↔	NA	NA	NA	NA	NA	T1 vs. T0 ↓ soft drink in both groups ↓ In > Cn (NS)
Leidy HJ, 2015 [30]	54 BS adolescents 18±1 yrs M/F, USA	Cn (BS) = 9 Normal-protein (NP) breakfast = 21 High-protein (HP) breakfast=24	NP and HP breakfasts	12 weeks	No food or drinks before 10:00 a.m.	NA	T0 vs. T1 HP vs. Cn: ↓ fat mass and % body fat	NA	↔	NA	NA	NA	NA	T0 vs. T1 HP vs. Cn: - ↓ daily food intake (-1724 ± 954 vs. +1556 ± 745 kJ) - ↓ fat consumption (-24 ±12 vs. -16±14 g) - ↓ daily hunger
Moore GF, 2013 [31]	111 primary schools (58 in socio-economically deprived areas) subjects aged 9–11 yrs T0: 4350 T1: 4472 M/F, Wales	Cluster-RCT Cn= 56 (T0: 2145; T1: 220) In= 55 (T0: 2205; T1: 2272)	School-based breakfast	1 year	BS: ≤2 days	NA	/	↓ BS in more deprived schools	NA	NA	NA	NA	NA	T1 ↑ healthy breakfast ↑ in more deprived Schools
Murphy S, 2010 [32]	111 primary schools (58 in socio-economically deprived areas) subjects aged 9–11 yrs T0: 4350 T1: 4472 M/F, Wales	Cluster-RCT Cn = 56 (T0: 2145; T1: 220) In = 55 (T0: 2205; T1: 2272)	School-based breakfast	1 year	BS: ≤2 days	NA	/	↔	NA	NA	NA	NA	NA	T1 ↑ healthy breakfast
Pbert L, 2016 [33]	126 adolescents grades 9–12th 8 public schools M/F, USA	Pair-matched cluster-RCT Cn = 58 In = 68	School nurse-delivered counseling	8 months	Number of days eat breakfast in past 7 days	CDC	T1 vs. T0 ↑ days/week eating breakfast in In vs. Cn (4.65 vs. 3.84 days)	↓	↔	NA	NA	NA	NA	T1 vs. T0 - ↔ fruit and vegetable intake, drinking soda, eating fast foods - ↔ barriers to healthy eating

**Legend:** BS = breakfast skipping; Cn = Control group; In = Intervention group; F = Females; M = Males; yrs = years; OB = obesity; OW = Overweight; BP = Blood Pressure; BMI = Body Mass Index; CDC = Center for Disease Control and Prevention; WHO = World Health Organisation; IOTF = International Obesity Task Force; CI = Confidence Interval; WG = whole grain; Recommended Energy Allowance = REA; ↑ = Increased; ↓ = Reduced; ↔ No variation; NA = Not available. The last seven columns describe the impact of BS on those variables.

**Table 3 nutrients-13-03331-t003:** Type of intervention and outcome measures in longitudinal studies and RCTs included in the systematic review.

Author, Year, Ref	Study Design	Intervention	Outcome Measures
Leatherdale ST, 2016 [18]	Longitudinal	School breakfast program: eating breakfast as part of a school program one or more days in a usual school-week (Monday to Friday)	COMPASS School Programs and Policies Questionnaire (SPP): paper-based survey completed annually by school administrators most knowledgeable about the school program and policy environment within a school
O’Dea JA, 2011 [19]	Longitudinal	National breakfast promotion campaign including classroom lessons about nutrition and ideas for a breakfast menu at each school canteen	Questionnaire collected demographic details of the students (gender, age, school grade/year, usual breakfast consumption patterns, breakfast consumption on the day of the study, and contents of breakfast consumed) Height and weight were measured
Ritchie LD, 2016 [20]	Longitudinal	School-based intervention: 3 breakfast policies: (1) breakfast in the cafeteria before the start of school (17 schools); (2) breakfast in the classroom (20 schools); (3) second chance breakfast (6 schools)	Student demographic data (sex, race/ethnicity, language spoken at home) obtained by survey completed by students in the classroom Diet quality measured by the Healthy Eating Index 2010
Traub M, 2018 [21]	Longitudinal	School-based health promotion program “Join the Healthy Boat”: training courses for teachers at primary school to promote healthy lifestyle choices in children, to increase physical, mental, and emotional abilities and, consequently, to attenuate the increase in body fat and thus to prevent overweight and obesity	Anthropometric data of the children (height, weight, waist circumference) were assessed in schools Parental questionnaires at baseline and follow-up (parent’s anthropometric data, child’s health behaviour, lifestyle, and socioeconomic background)
Whitaker RC, 2019 [22]	Longitudinal	2 district-wide school start time (SST) changes implemented: (1) a 50 min delay, from 7:20 a.m. to 8:10 a.m., in high schools (grades 9–12) and secondary schools (grades 7–12) and (2) a 30 min advance, from 8 a.m. to 7:30 a.m., in middle schools (grades 7–8)	Repeated cross-sectional school surveys to evaluate the impacts of SST changes in a US school district Self-report surveys in the areas of mood, self-regulation, safety, and health
Ask AS, 2006 [23]	RCT	Intervention group: consisted of served breakfast at the beginning of each school day; students were also offered a food supplement (vitamins, minerals, and omega-3 fatty acids) Control group: was not served breakfast but got the same information about the importance of a healthy diet All parents were encouraged to provide a packed lunch for their children every day	Food frequency questionnaire (covered frequency intake of 27 food items commonly used in the Norwegian diet) Height and weight were measured with standard equipment by the school nurse before and after the study Diet quality measured by a healthy eating index
Au LE, 2015 [24]	RCT	Online vs. in-person nutrition education on the importance of having breakfast	Participants (in-person and online groups) completed a questionnaire to assess breakfast knowledge, attitudes, and behaviors
Bauer KW, 2020 [25]	RCT	Intervention group: Breakfast in the Classroom (BIC) + breakfast-specific nutrition education (18 lessons), social marketing to promote consumption of a healthy breakfast, marketing at 14 corner stores promoting healthier breakfast foods and drinks, parent outreach via family newsletters (24 newsletters) and information available at schools’ parent events Control group: offered breakfast in the cafeteria before school	Breakfast Patterns Survey (BPS) at baseline, midpoint, and endpoint to measure students’ food and drink consumption Data on students’ background were obtained from the school district Weight and height were measured, and BMI calculated
Christensen CB, 2019 [26]	RCT	Breakfast club intervention (BCI) based on wholegrain (WG) products Intervention group: schools served a free wholegrain breakfast every school day as part of the first lesson, either in the classroom or in the cafeteria; the breakfast consisted of a choice among four WG cereal products Control group: schools carried on as normal without the availability of free breakfast	Questionnaires measured dietary intake (breakfast intake, type of breakfast, frequency of snacking unhealthy food) and background (demographic information, behavioral involvement, and attitudinal questions)
Crepinsek MK, 2006 [27]	RCT	Intervention group: universal-free school breakfast for 3 consecutive school years Control group: traditional means-tested School Breakfast Program	24-h dietary recall to evaluated food intake (taken with students only and with students and parents together) A subsample of students completed a second dietary recall 7–10 days after the first
Elseifi OS, 2020 [28]	RCT	Intervention group: nutritional education message based on Pender’s health promotion model: a specific 58-item questionnaire to assess behavioural factors related to breakfast consumption; 5 sessions (one session every week), 30 min each Control group: any education message	General self-administrated questionnaires (sociodemographic characters, quality and frequency of breakfast intake) Questionnaire for the Pender’s Health Promotion Model (HPM) regarding Breakfast Consumption Weight, height, BMI, and blood pressure were measured by researcher
Kobel S, 2014 [29]	RCT	School-based health promotion program “Join the Healthy Boat” Intervention group: training courses for teachers at primary school to promote healthy lifestyle choices in children, to increase physical, mental, and emotional abilities, and, consequently, to attenuate the increase in body fat and thus to prevent overweight and obesity Control group: followed the regular school curriculum	Parental questionnaire evaluated parameters: daily screen media time, physical activity behavior, soft drink consumption and breakfast patterns, parental education levels, height, and body weight Child’s anthropometric measurements (height and weight) were taken by trained technicians and BMI was calculated
Leidy HJ, 2015 [30]	RCT	Intervention group: Normal protein (NP; 13 g protein) and High protein (HP; 35 g protein) groups provided with specific breakfast meals to consume between 6:00 and 9:45 a.m. each day Control group: continued to skip breakfast (with nothing to eat/drink, besides water) before 10:00 a.m.	Body weight, body composition, 3-day free-living perceived appetite, and 3-day daily food intake (24-h dietary recalls) were assessed at baseline and 12 weeks
Murphy S, 2010 [31]	RCT	Intervention group: school-based breakfast before the commencement of classes, without any cost being borne by parents; aim of intervention was to encourage breakfast but also improve the nutritional quality of children’s breakfasts Control group: schools in control group were asked to refrain from setting up a breakfast scheme during the 12-month evaluation period	Data were collected by attitudes and dietary recall questionnaires (modified version of the Day in the Life Questionnaire, Strengths and Difficulties Questionnaire)
Moore GF, 2013 [32]	RCT	Intervention group: school-based breakfast before the commencement of classes, without any cost being borne by parents; aim of intervention was to encourage breakfast but also improve the nutritional quality of children’s breakfasts Control group: schools in control group were asked to refrain from setting up a breakfast scheme during the 12-month evaluation period	Data were collected by attitudes and dietary recall questionnaires (modified version of the Day in the Life Questionnaire, Strengths and Difficulties Questionnaire)
Pbert L, 2016 [33]	RCT	Intervention group: “Lookin’ Good Feelin’ Good”: a school nurse-delivered counseling intervention (6 weekly 30-min individual sessions + maintenance phase of 6 monthly sessions), an after-school exercise program (3 sessions/week for 8 months); program goals/focus of the counseling: eating healthy and being more physically active Control group: 12-session nurse contact with weight management information to reduce BMI and improve diet and activity among overweight and obese adolescents	Dietary intake was assessed with a 24-h dietary recall interview—Interactive Nutrition Data System Physical activity and nutrition behaviors were monitored with questionnaires Height, weight, and waist circumference were measured

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
