# Peer review of "Breakfast Skipping, Weight, Cardiometabolic Risk, and Nutrition Quality in Children and Adolescents: A Systematic Review of Randomized Controlled and Intervention Longitudinal Trials"

_nutrients, 2021, doi:10.3390/nu13103331_

Round 1
Reviewer 1 Report
The paper entitled ‘Breakfast skipping, weight, cardiometabolic risk, and nutrition quality in children and adolescents: a systematic review of randomized controlled and intervention longitudinal trials’ aims to review problem of overweight and obesity and relation of the issue with breakfast skipping. The paper is interesting and reviewed up-to-date scientific news nevertheless some points have to be improved/corrected:
- Abstract is too long. Please reduce this part.
- Keywords: max number of keywords is 10. Please remove excess of keywords
- Table 1. Is hard to read. There is much information but analysis of them in present form is difficult. Please rebuild the table or present information in other form.
On the whole, the paper is well prepared. Discussion is sufficient, but presented studies results should be analyse more detailed.
Reviewer 2 Report
Ref: nutrients-1351711
Title: Breakfast skipping, weight, cardiometabolic risk, and nutrition quality in children and adolescents: a systematic review of randomized controlled and intervention longitudinal trials
Recommendation: Major review
Comments:
- This type of paper should cover more than 10 manuscripts. The Authors excluded 441 only based on the title. I would recommend to expand the paper and to thoroughly study the subject of rejected papers.
- In my opinion, nicely described and informative schemes would be an additional value.
Round 2
Reviewer 1 Report
Dear Authors,
Thank you for manuscript improvement in accordance with my suggestions. Present form of the paper is more readable and informative. I accept the paper to publish.
Reviewer 2 Report
In my opinion (after small changes after reviews), the paper is ready for publication.